# Fine-Tuning through Generations: Advances in Structure and Production of CAR-T Therapy

**DOI:** 10.3390/cancers15133476

**Published:** 2023-07-03

**Authors:** Zhibo Zheng, Siyuan Li, Mohan Liu, Chuyan Chen, Lu Zhang, Daobin Zhou

**Affiliations:** 1Department of International Medical Services, Peking Union Medical College Hospital, Chinese Academy of Medical Sciences and Peking Union Medical College, Beijing 100730, China; 2Department of Hematology, Peking Union Medical College Hospital, Chinese Academy of Medical Sciences and Peking Union Medical College, Beijing 100730, China; 3Department of Breast Surgery, Peking Union Medical College Hospital, Chinese Academy of Medical Sciences and Peking Union Medical College, Beijing 100730, China; 4Department of Gastroenterology, National Clinical Research Center for Digestive Disease, Beijing Digestive Disease Center, Beijing Friendship Hospital, Capital Medical University, Beijing 100730, China

**Keywords:** chimeric antigen receptor T-cells, evolution, structure, production

## Abstract

**Simple Summary:**

Chimeric antigen receptor (CAR)-T cell therapy has been highly successful in treating hematological malignancies like leukemia and lymphoma, leading to advancements in immunotherapy. The approval of anti-CD19 CAR-T cells by the FDA in 2017 has paved the way for CAR-T therapies to be extended to various cancers, autoimmune disorders, viral infections, and fibrosis. However, the therapy faces challenges, including severe side effects like cytokine release syndrome and graft-versus-host disease. Additionally, CAR-T therapy has limited efficacy against solid tumors and antigen escape, poor persistence, and difficulties in industrialization. Overall, this review provides a comprehensive understanding of the challenges and opportunities associated with CAR-T cell therapy, highlighting the need for further advancements in CAR-T cell construction and production. By facilitating future research and development, CAR-T therapy has the potential to become a powerful tool in the fight against cancer and other diseases.

**Abstract:**

Chimeric antigen receptor (CAR)-T cell therapy is a promising form of immunotherapy that has seen significant advancements in the past few decades. It involves genetically modifying T cells to target cancer cells expressing specific antigens, providing a novel approach to treating various types of cancer. However, the initial success of first-generation CAR-T cells was limited due to inadequate proliferation and undesirable outcomes. Nonetheless, significant progress has been made in CAR-T cell engineering, leading to the development of the latest fifth-generation CAR-T cells that can target multiple antigens and overcome individual limitations. Despite these advancements, some shortcomings prevent the widespread use of CAR-T therapy, including life-threatening toxicities, T-cell exhaustion, and inadequate infiltration for solid tumors. Researchers have made considerable efforts to address these issues by developing new strategies for improving CAR-T cell function and reducing toxicities. This review provides an overview of the path of CAR-T cell development and highlights some of the prominent advances in its structure and manufacturing process, which include the strategies to improve antigen recognition, enhance T-cell activation and persistence, and overcome immune escape. Finally, the review briefly covers other immune cells for cancer therapy and ends with the discussion on the broad prospects of CAR-T in the treatment of various diseases, not just hematological tumors, and the challenges that need to be addressed for the widespread clinical application of CAR-T cell therapies.

## 1. Introduction

Chimeric antigen receptor (CAR)-T cell treatment has revealed outstanding clinical success in treating hematological malignancies as leukemia as well as lymphoma during the past few years, pathing the way toward new developments in immunotherapy. The achievements of anti-CD19 CAR-T cells in curing B-cell malignancies are unprecedented, leading to their approval by the U.S. Food and Drug Administration (FDA) in 2017 [1,2,3,4]. So far, CAR-T therapies have been extended to focus on a wide range of cancers, autoimmune disorders, viral infections, and fibrosis. Although attractive results in clinical trials have been reported, CAR-T cell therapy still faces significant challenges, with severe side effects being a major obstacle that urgently needs to be addressed, including cytokine release syndrome (CRS), graft-versus-host disease (GVHD), tumor lysis syndrome (TLS), and immune effector cell associated neurotoxicity syndrome (ICANS) [5,6,7]. Among which, severe CRS requiring management in the intensive care unit and treatment with tocilizumab or corticosteroids developed in 4/32 NHL patients treated with CD19 CAR-T cells [8]. Other challenges such as antigen escape, limited persistence, poor efficacy against solid tumors, and difficulty in industrialization also [5] pose limitations to the use of CAR-T cell therapy [9,10,11]. Recent studies revealed that appropriate CARs’ composition and structural design could minimize off-target effects and enhance tumor eradication [12,13,14]. Subtle modifications to the CAR-T manufacturing procedure can also be a pivotal determinant of function. Nonetheless, the association between those factors is sophisticated, yet no general design principles are available for predicting the in vivo function.

The aim of this review is to provide a comprehensive understanding of the current state and future developments in CAR-T cell construction and production. Our objectives are to examine how subtle modifications in CAR-T structure and production can improve therapy, step by step and generation by generation. To achieve this, we will begin by discussing the evolution of CAR-T cell therapy and providing a brief introduction to the fundamental structure of CAR-T cells. We will also examine recent findings on the influence of each component on engineered cell signaling and function, explore the manufacturing procedure of CAR-T cells, and highlight the need for further optimization. In addition, we will address the use of other immune cells in cancer therapy and the challenges associated with solid tumors and non-neoplastic disorders. By achieving these objectives, we hope to facilitate future advancements in CAR-T cell construction and production and provide physicians with a thorough understanding of the challenges and opportunities associated with CAR-T cell therapy.

## 2. Evolution of CAR-T Cells

Since the concept of the CAR was first introduced in 1989, CAR-T therapies have been through five generations of development (Figure 1) as a result of developments in the construction of the CAR. Through design innovation and structural optimization, quite a few preclinical studies have emerged demonstrating promising efficacy and safety, greatly expanding the clinical application of CAR-T therapy [15,16].

First-generation CARs are fusion proteins that typically consist of a single-chain variable fragment (scFv) derived from an antibody, which serves as the extracellular antigen-binding domain, linked to an intracellular signaling domain usually containing the CD3ζ chain of the T-cell receptor (TCR). In second-generation CARs, the addition of a costimulatory domain fused to CD3ζ, such as CD28 or 4-1BB, enhances CAR-T cell activity by promoting the expansion and persistence of the engineered cells in vivo. Third-generation CARs have been developed with multiple costimulatory domains to further enhance T-cell activation, proliferation, and cytokine production. Fourth-generation CAR-T cells are designed to secrete a specific cytokine upon recognizing the target antigen, which is accomplished by adding a second gene to the CAR construct that encodes for a cytokine or another immune modulatory molecule (enzymes or ligands). The fifth-generation CARs are being designed as universal CARs including SUPRA CAR and BBIR CAR. SUPRA CAR is a two-component receptor system comprising a universal receptor (zipCAR) expressed on T cells and a separate scFv (zipFv) molecule targeting specific antigens. BBIR CAR is a biotin-binding immune receptor consisting of an avidin extracellular domain linked to an intracellular T-cell signaling domain.

### 2.1. First Generation

In 1988, Rosenberg et al. first reported the clinical data among 20 metastatic melanoma patients using tumor-infiltrating lymphocytes therapy [17]. The remission rate in melanoma patients who had never received IL-2 previously reached 60% (nine out of 15 patients) compared to the 40% who had earlier received IL-2 (two out of five patients). This extraordinary discovery prompted researchers to consider if it is feasible to genetically design T cells with receptors that could detect a specific antigen independent of a major histocompatibility complex (MHC) expression. Thus, the concept of the CAR was pioneered by Eshhar et al., who first fused the various regions of the heavy and light chains of a monoclonal antibody to the constant regions of the T-cell receptor (TCR), demonstrating that the synthetic receptors could identify tumor antigens [18]. Shortly after the initial discovery, the structure of the first generation of CAR-T cells was established (Figure 1), including an extracellular domain for antigen recognition, an intracellular signaling domain mediating T-cell activation, and a transmembrane structural domain for optimal cell activation through CARs’ dimerization and interactions with endogenous TCRs [19].

Even though it was safe to administer large amounts of gene-modified tumor-reactive CAR-T cells to patients, the clinical value of the first-generation CAR-T cells was constrained by their subpar proliferation and poor persistence [20,21]. Moreover, the first-generation CAR-T therapy using T cells transfected with single-chain receptors might have benefitted considerably from the accompanying administration of cytokines due to insufficient secreted cytokines [22].

### 2.2. Second Generation

Naïve T cells require dual-signaling stimulation to become effector T cells after exiting the thymus into the peripheral blood. The first signal is initiated by the TCR recognizing the antigen peptide-MHC complex (p-MHC) on antigen-presenting cells’ (APCs) surface, while the interaction of costimulatory molecules between T cells and APCs induces the second signal. Therefore, first-generation CARs that included only CD3ζ sequences were not capable of activating CAR-T cells in the absence of costimulatory signals. Accordingly, Maher et al. first found out that human primary T lymphocytes expressing a fusion receptor directed to a prostate-specific membrane antigen (PSMA) and containing TCRξ and CD28 signaling elements could efficiently lyse PSMA-expressing tumor cells [23]. Thus, the second-generation CAR-T cells were created with the addition of costimulatory domains primarily derived from the CD28, 4-1BB, OX40, or CD27.

Through the coordination of intracellular signals and costimulatory molecules (Figure 1), the second generation of the CAR showed improved proliferation, cytotoxicity, and sustained response of CAR-T cells in vivo. A direct comparison of T cells containing costimulatory domains versus those with only one signaling domain showed that CAR-T cells containing the CD28 internal domain had a significantly increased expansion and persistence [24]. Furthermore, early clinical trials in patients with 17p-deficient refractory chronic lymphocytic leukemia reported a marked disappearance of leukemic cells and late progress of tumor lysis syndrome after treatment with CD19 CAR-T cells linked to CD3ξ and 4-1BB signaling domains [25,26,27].

### 2.3. Third Generation

To better activate T cells, another subtle modification was made on the basis of second-generation CAR to generate ‘third-generation CAR’. The third-generation CAR was engineered by integrating an extra co-stimulatory structural domain in the second-generation one, typically consisting of CD28/4-1BB/CD3ξ or CD28/OX40/CD3ξ in the intracellular domain (Figure 1). Early clinical trials in patients with leukemia and non-Hodgkin lymphoma (NHL) reported superior expansion and longer persistence, especially for those with a milder tumor burden and lower levels of normal B cells [28,29]. However, Ramello et al. suggested that, compared to third-generation CARs, second-generation CARs could activate more sources of CD3ζ signaling [30], thus inducing more intense signaling and superior antitumor efficacy. Considering the inconsistent trial results as well as the limited number of patients who received the third-generation CAR-T treatment, whether more co-stimulatory molecules are related to better CAR-T efficacy has not been conclusively established.

### 2.4. Fourth Generation

Clinical research has indicated that CAR-T cell therapies of the second and third generation have given rise to breakthroughs in treating hematological malignancies, with long survival periods ensuring long-term surveillance and clearance of tumor cells. However, solid-tumor treatment encountered significant challenges, mainly because of inadequate T-cell penetration and an unfavorable tumor microenvironment [31,32]. Meanwhile, the occurrence of adverse events emphasizes the necessity of improved safety protocols for CAR-T cells [33,34], driving the subtle modifications in the design of fourth-generation CARs which contains a controlled switch, a suicide gene, or an element that enhances T-cell function [35,36]. For example, the incorporation of an inducible caspase-9 gene element can reduce toxicity by apoptosis of the CAR-T cells [37,38]. More recently, Louai Labanieh et al. developed SNIP, a protease-based, drug-modulated platform to control CAR-T cell activation, showing greater potency than constitutive CARs without leaky activity (Figure 1) [39].

### 2.5. Fifth Generation

Normally, CAR-T cell therapy works by expanding a patient’s own tumor-specific T cells in vitro and then reinfusing them into the same patient, which is an extremely personalized method with high cost. The concept of the ‘universal CAR’ is attractive for the possibility of large-scale production which could simultaneously reduce the costs of the treatment. Universal CARs use two ‘third-party’ systems, BBIR (biotin-binding immune receptor) CAR or SUPRA (split, universal, programmable) CAR (Figure 1), to split the extracellular antigen-targeting structural domains and T-cell signaling units to confer CAR-T cells the ability to recognize multiple antigens [40,41]. Meanwhile, T cells obtained from healthy allogeneic donors are genetically edited (ZFN, TALEN, and CRISPR/Cas9) to disrupt TCR genes and HLA class I genes for graft-versus-host disease (GVHD) prevention. Although such general-purpose CAR-T cell therapy has encountered highly technical barriers, research on universal CAR-T has come a long way.

To sum up, subtle modifications of the structures of CARs, generation by generation, have brought huge differences to CAR-T therapies and stirred a huge amount of passion for cancer treatment. The first generation had weak anti-tumor effects, while the third and fourth generations have been strengthened along with increased toxicity and other side effects. Nowadays, second-generation CAR-T cells with intermediate efficacy dominate in current clinical scenarios and the FDA-approved CAR-T cell therapy has increased year by year to surge ahead in the cellular immunotherapy revolution (Table 1). Moreover, CAR-T cell therapies are making remarkable progress in several preclinical and early-phase clinical trials and are currently advancing at a rapid pace (Table 2).

## 3. Structure Modification of CAR

CAR-T cell therapy is a groundbreaking advancement that combines the antigen-specific properties of the CAR with the powerful intrinsic cytotoxic capacity of T cells. CARs, no matter which generation, comprise four fundamental parts: an antigen-binding domain, a hinge, a transmembrane domain, and an intracellular signaling domain. Even though CAR-T cell treatment is innovative with powerful and long-lasting anticancer activity, significant adverse effects, and a lack of efficacy in solid tumors have gradually emerged as challenges. As a result, numerous studies are looking at enhancing the functionality of CAR-T cells by adjusting each module structure, potentially affecting efficiency and toxicity [86,87].

### 3.1. Antigen Recognition and Binding Domains

The antigen-binding domain located in the extracellular region of the CAR is the basis for the CAR-specific binding of tumor antigens. It mainly involves monoclonal antibodies with a variable heavy chain (V_H_) and a variable light chain (V_L_), linked by a flexible linker to form a single chain variable fragment (scFv) [88]. (Gly4Ser)3 peptide is the most common linker, which takes advantage of the flexibility of glycine residues and the solubility of serine residues [89]. According to recent research, in contrast to the shorter linker forming homodimers, the longer linker CAR generated monomers in solution, connected to elevated tonic signaling and superior anti-tumor function [90,91,92]. Certain scFv properties, such as immunogenicity, affinity, and antigen epitope, significantly impact CAR function besides identifying and binding the target antigen.

#### 3.1.1. Non-Immunogenic scFv

The typical scFv sequences are derived from murine, and some patients showed limited persistence and a high risk of relapse, which was mediated by the anti-mouse reactivity [93]. Therefore, revolutionized CAR-T cells harboring a humanized tumor-reactive scFv composition were developed to render components less immunogenic. Standard methods include replacing amino acids in the scFv framework of a mouse-based CAR with its human counterparts or incorporating fully human antibody fragments into CAR constructs [14,94,95]. Specifically, CARs with human-derived heavy-chain-only variable fragments are associated with a lower risk of immunogenicity which is attributed to the deletion of linkers. Clinical trials have demonstrated that anti-CD19 CAR-T cells with entirely human binding domains are capable of mediating durable disease remission with a long-lasting existence in both children and young adults with relapsed or refractory (R/R) B-acute lymphoblastic leukemia (B-ALL), even in patients who have previously received prior but unsuccessful CAR-T cell therapy [96,97,98]. Nevertheless, due to the variable disease burden and inconsistent treatment duration, the assumption of decreased immunogenicity of humanized CARs was not solidly verified in direct comparisons between human-derived and murine products. Larger-scale clinical trials are needed to understand the factors influencing humanized CAR-T cells’ effectiveness.

Aside from scFv-based CARs, the non-antibody-based methods can target tumor cells to interact between natural receptors and their ligands. Since human peptide sequences are more easily recognized as autologous proteins than conventional scFvs, tumor-specific receptor-ligand interactions are expected to become less immunogenic. For example, with ErBB-targeted CAR-T cells, 6 out of 10 head and neck squamous cell carcinoma patients achieved disease control with no reported dose-limiting toxicity [99]. Initial clinical data also showed that intracranial infusions of IL13Rα2-targeted CAR-T cells had no relation to any grade 3 or higher toxic effects in patients with recurrent glioblastoma [100]. Other ligand-based CARs included follicle-stimulating hormone (FSH) CARs targeting the FSH receptor expressed on ovarian cancer cells [101], adnectins binding to the epidermal growth factor receptor (EGFR), together with the granulocyte-macrophage colony-stimulating factor (GM-CSF) relating to the GM-CSF receptor (CD116) [102].

#### 3.1.2. Affinity-Tuned scFv

The affinity of scFv fundamentally determines the function of a CAR for the target antigen, and both low and high affinity may allow for a delicate control. The affinity of scFv is expected to be high enough to identify tumor cells and activate T cells whereas excessive affinity can result in activation-induced cell death (AICD) and potential toxicities. Conversely, it would be anticipated that diminished antigen affinity could avoid targeting healthy tissues with minimal antigen. In a recent comparison of CARs bearing different affinity scFv binding to a similar epitope and cross-reactive with murine oncofetal antigen glypican 3 (GPC3), Torchia et al. demonstrated that CAR-T cells with high affinity were toxic in vivo. In contrast, low-affinity CAR-T cells retained cytotoxic function against antigen-positive tumor cells without exhibiting toxicity to normal tissues [103]. A new CD19 CAR-T, the high-affinity binder employed in several clinical researches [4,104], illustrated elevated proliferation and cytotoxicity in vitro than FMC63 CAR-T cells. Similar findings can be seen in CAR-T cells targeting CD19, CD38, and HER2 [2,4,93,104,105], collectively suggesting that an appropriate reduction in the scFv affinity may not significantly affect the efficacy of CAR-T cell therapy and could also reduce the occurrence of off-tumor toxicity [105,106,107,108,109]. As a result, the affinities should be balanced between the optimized strength of the anti-tumor response as well as the possible risk of on- and off-target toxicity.

In addition, by increasing the complexity of the CAR design, synthetic biology-based strategies can exert an impact on the scFv affinity. The antigen density-sensitive switch is designed to detect the HER2 antigen expression using a syn-Notch receptor [110], a filter that allows transcriptional induction only to occur when T cells encounter targets expressing high antigen levels. Once it has been successfully overcome, the induced high-affinity CAR can promote the growth of T cells and exert their tumor cell-killing effect only at the site of high antigen-density tumor cells with no harmful impact on healthy cells. Inspired by these new techniques, the therapeutic window of CAR-T therapy has been widened since more tumor-associated antigens can be selected as targets.

#### 3.1.3. scFv Targeting Multiple Antigens

CAR-T cells need to specifically recognize tumor surface antigens to ensure that they target only tumor cells and do not harm normal tissue. However, studies have shown that tumors can evade immune recognition by reducing or losing surface antigens [6,93]. In solid tumors, CAR-T has struggled to identify highly specific, safe, and non-off-target therapeutic targets due to tumor heterogeneity and a protective tumor microenvironment that can shield tumors from CAR-T cells [111].

There are two strategies to optimize scFvs for targeting multiple antigens. The first strategy involves co-transducing two separate vectors that encode different CARs into a single T cell, or constructing a bispecific receptor with two separate chimeric antigen receptors on each cell [111,112]. Preliminary results from clinical trials using dual-targeted CAR-T cells (CD19/CD22 or CD19/BCMA) have demonstrated promising efficacy in hematological malignancies [113]. Similarly, in solid tumors, dual targeting resulted in superior anti-tumor responses compared to a single targeted therapy. It is important to select target antigens that not only improve antitumor response but also decrease antigen escape mechanisms to prevent relapse. The second strategy uses tandem bispecific CARs, in which the extracellular domains of the two binding domains can recognize either antigen and initiate effector functions. Choi et al. developed a CAR-T cell that secretes an EGFR/CD3 BiTE and targets EGFRvIII to eradicate glioblastoma [10]. This approach was validated in a mouse model, where 80% of the mice achieved complete remission without any detectable tumor after three weeks of treatment.

#### 3.1.4. Target Epitope

Another essential part of engineering scFv is the location of the target antigen-binding site. When two scFvs targeting the same epitope were compared in the setting of CD22-CARs, it was discovered that an alternate CD22 binder (m971) with a lower affinity was most effective, probably because the targeted epitope was more accessible [8,28]. CARs targeting the proximal region of the mesothelin molecule membrane expressed higher levels of cytotoxicity and cytokine secretion than those targeting the distal epitope of the membrane. Researchers hypothesized that the rigid structure of the membrane-proximal region could exhibit robust signal transduction. Furthermore, mesothelin’s membrane-side region has functional interactions with proteins such as CA125, which may impede CAR binding [41]. Thus, in addition to steric availability, structural and functional characteristics of the target epitope are also highly indispensable in CAR design.

### 3.2. Hinge and Transmembrane Domains

The hinge and transmembrane domains of CARs are essential for linking the extracellular antigen-binding structural domain to the intracellular signaling structures. Amino acid sequences from CD8, CD28, IgG1, or IgG4 were used in CAR hinge domains, providing sufficient flexibility and adequate length. They could be used alone or in conjunction with a spacer sequence that projects the CAR to access membrane proximal antigen epitopes [114,115].

It is worth noting that the variations in the hinge length could dramatically enhance CAR antigen binding and signaling, which is primarily based on the position of the target epitope and the accessibility. Longer spacers offer enhanced flexibility and allow more efficient access to the membrane-proximal epitope, while the shorter hinges provide better activation of CAR-T cells. On the one hand, the activity of CD22-specific CAR could be enhanced with a longer distance between the T cell surface and the CD22 epitope via a CH2CH3 domain. Similarly, anti-5T4 and anti-NCAM chimeric immune receptors could only display a superior effect of cytokine release and cytotoxicity with a long spacer. On the other hand, researchers on the 2A2 ROR1-CAR found that intermediate (hinge-CH3) and short (hinge-only) spacers were superior in T cell cytokine secretion and proliferation after recognition of tumor cells compared to long (hinge-CH2-CH3) spacers [90].

In addition to hinge length, considerable efforts are now being spent to optimize the hinge compositions. CD19-CAR constructed with a long spacer of IgG4 hinge-CH2-CH3 is associated with a lack of antitumor activity in vivo, as it interacts with Fcγ receptors (FcγRs) [91]. However, the diminished persistence and anti-tumor activity can be restored by altering specific regions of the CH2 domain required for Fc receptor binding. Hinge modification was also accomplished in the attempts for the treatment of AML. Leick et al. engineered the CD8 hinge and transmembrane-modified CD70 CAR-T cells to alleviate cleavage of the extracellular portion of CD27 and demonstrated that the hinge itself displays an additional role in reducing cleavage [116].

Transmembrane domains, frequently derived from type I proteins such as CD3, CD28, CD4, or CD8α, affect CAR-T cells’ activity and stability. CARs containing the CD3ζ transmembrane domain mediate CAR dimerization and incorporation into endogenous TCRs, which may enable CAR-mediated T-cell activation [71,117]. Interestingly, CARs comprising the CD28 transmembrane domain are also proved to dimerize and simultaneously are more stable than those harboring the transmembrane region of CD3ζ [118]. Moreover, the transmembrane domain may contribute to the recycling and stability of the CAR in the membrane, influencing CAR expression levels directly [119].

As for the comparison between CARs with hinge and transmembrane regions from either CD8α or CD28, current research also demonstrated that the CD28 hinge reduced the antigen density threshold for CAR activation. Therefore, anti-CD19 CAR-T cells with hinge and transmembrane regions from CD28 produced higher cytokines levels and exhibited high AICD levels than those with CD8α hinge and transmembrane domains [120]. However, a direct comparison of CD22-targeting CAR bearing a CD8α hinge or a CD28 hinge showed that CD8α exhibited more significant cytotoxicity against a decreased antigen density CD22 positive leukemia [121]. Further exploration into the biophysical and dynamic characteristics of CD8 revealed that proline-enabled dynamic switching and local structural features with disulfide bridging between dimeric molecules raise signaling within the reduced antigen context. In the cell membrane context, the isolated hinge domain may function differently from the full-length CAR, and interactions with the cell surface may stabilize specific local structures and/or modify exchange dynamics. More recently, Leick et al. sought to improve CAR-T treatment effectiveness in acute myeloid leukemia and designed a series of variants targeting different components of CAR. Among these, CD8 hinge and transmembrane-modified CD70 CAR-T cells were less susceptible to cleavage with a stronger binding avidity and increased expandability [116].

### 3.3. Intracellular Signaling Domains

The intracellular signaling domain is another important component of the CAR, which typically contains an activation domain and one or more co-stimulatory domains. The activation domain is usually the T cell receptor TCR/CD3ζ chain or the immunoglobulin Fc receptor FcεRIγ chain, which contains immunoreceptor tyrosine activation patterns (ITAMs) and plays a role in T-cell signaling [122]. Nevertheless, signaling mediated by these motifs alone is not sufficient for generating productive T-cell responses. Therefore, modified second-generation and later carts usually comprise one or more co-stimulatory structural domains from the CD28 receptor family (CD28, ICOS) or the tumor necrosis factor receptor family (4-1BB, OX40, CD27). By coordinating the dual activation of co-stimulatory molecules and intracellular signaling, T cells continue to proliferate and release cytokines, enhancing the anti-tumor capacity.

The number reduction of functional ITAMs in a CAR molecule has been demonstrated to have decreased T-cell signal strength and downstream effector functions [123]. With only one of three CD3ζ ITAMs, the CD19-CD28ζ CAR-T cells were engineered to increase persistence and lower the level of T-cell exhaustion [124]. Moreover, ITAM mutants could calibrate CD28-based CAR-T cells’ activation, through which CAR-T cells can be guided to various fates, allowing a CD28-based CAR-T cell with mutations in ITAM1 and ITAM2 domains to maintain long-lasting memory but retain efficient anti-tumor function [125].

CAR-T cells with distinct costimulatory domains showed variable dynamics, providing another area that can be altered depending on the tumor type, tumor burden, the density of the antigen, and other safety concerns [126]. The most commonly used co-stimulatory domains are CD28 and 4-1BB. As one of the members in the immunoglobin superfamily, CD28 is a typical T-cell costimulatory receptor, which can compete with the co-inhibitory receptor CTLA4 and bind to B7 molecules CD80 and CD86 on APCs. While some co-stimulatory molecules are only expressed when T cells are activated, CD28 is expressed in both quiescent and recently activated T cells. By enhancing transcription and mRNA stability, the cytoplasmic motif YMNM of CD28 binds and activates phosphatidylinositol 3-kinase (PI3K) and connexin Grab-2, increasing T-cell proliferation and IL-2 production [127]. When bound to the 4-1BBL ligand, 4-1BB leads to the recruitment of TRAF1, TRAF2, and TRAF3 to the intracellular domain and results in the formation of a 4-1BB signaling complex consisting of multiple proteins including kinases and ubiquitin ligases. Ultimately, it activates signaling pathways such as NF-kB and PI3K-AKT [128], inducing the proliferation of cytotoxic T cells, secreting pro-inflammatory cytokines, and expanding effector and memory T cells.

Previous studies have shown that the 4-1BB costimulatory domain is slower to expand than the CD28 costimulatory domain but more persistent, although there is no conclusive evidence in clinical studies on the superiority between the two domains [106,124,125,126]. Attempts to uncover mechanisms demonstrated that T cells expressing CD28-costimulation exhibited higher levels of cytokine, especially for the elevated secretion of IL-2, promoting T cell proliferation [129,130,131,132]. In addition, CAR-T cells with a 4-1BB co-stimulation can weaken the T-reg cells’ inhibitory effect, along with decreased expression of IL-10 and TGF-b. Furthermore, CD28-based CAR-T cells have a more effector-like memory phenotype and a higher level of glycolytic metabolism, which is connected to rapid activation and more pronounced changes in protein phosphorylation. Nevertheless, CAR-T cells incorporating 4-1BB as a co-stimulatory domain have a more central memory phenotype with a lower degree of phosphorylation and rely on fatty acid metabolism [133,134,135]. To specifically study signal transduction by CARs, Philipson et al. demonstrated the necessary and nonredundant role of NF-κB signaling in promoting the survival of 4-1BB CAR-T cells [136].

Though not yet been tried on patients, T cells carrying CARs that contain alternative co-stimulatory domains have shown effectiveness in preclinical models. OX40, CD27, CD40, HVEM, and GITR, which are all members of the tumor necrosis factor receptor superfamily (TNFRSF), have become well-known as co-stimulatory domains in CAR-T cell treatment [137,138,139,140,141]. Recently, Lai et al. found that the novel costimulatory domain C3aR dramatically improved CAR-T therapeutic efficiency in hematological malignancies, with long-term effects through memory T-cell production as well as Th17 expansion [142].

Overall, these results showed that it is essential to carefully analyze each structural domain’s function in a specific CAR/antigen interaction. Even if they do not directly interact with antigens, CAR-driven T-cell performance can be significantly impacted by areas. Equipped with the above modifications in the constructure, the CAR-T cells then mediated anti-tumoral effects through the perforin and granzyme, Fas and Fas-ligand interaction, as well as the release of cytokines to sensitize the cancer stroma (Figure 2).

## 4. Structure Modification of T Cells

TCRs are proteins present on the surface of T cells that recognize peptide antigens displayed by MHC molecules on the surfaces of other cells. However, since TCRs may recognize self-antigens or non-specific antigens, they can lead to off-target effects and potentially harmful immune responses. Therefore, the removal of endogenous TCRs and the elimination of MHC expression are key strategies to prevent off-target toxicity in CAR-T cell therapy.

The three most widely used gene editing techniques for selectively targeting and cleaving genes that encode TCRs are zinc finger nucleases (ZFN), transcription activator-like effector nucleases (TALEN), and the CRISPR/Cas9 system [36,143,144,145]. Alternatively, TCR-negative T cells, such as TCR-deficient T cells or T cells from TCR-transgenic mice, can be used as starting material for CAR-T cell therapy [146]. These T cells have either naturally occurring or genetically engineered TCR gene deletions that result in a lack of TCR expression on their surface. Non-genetic approaches that selectively deplete TCR-positive T cells include the use of monoclonal antibodies against the TCR, which selectively bind and eliminate TCR-positive T cells [147]. Drug-induced suicide gene systems, such as the herpes simplex virus thymidine kinase (HSV-TK) system, can also specifically eliminate TCR-positive T cells when administered with prodrugs such as ganciclovir [148]. These approaches have the potential to improve the safety and efficacy of CAR-T cell therapies.

Various strategies have been employed to generate CAR-T cells with reduced MHC antigen expression, enabling treatment across HLA differences. Direct editing of HLA patterns or the disruption of the β2-microglobulin (B2M) gene encoding MHC class I (MHC I) can eliminate allogeneic rejection of CD8+ T cells [36,149]. However, CD4+ T cell-mediated rejection of CAR-T cells may still occur due to T-cell activation, resulting in upregulation of MHC II. To mitigate this rejection, the MHC II gene of the master transcription factor CIITA can be deleted to eliminate the expression of MHC II [150]. Recently, Jo et al. developed an immunorepulsive universal CART cell scaffold that lacks TCRαβ and HLA class I but features surface-exposed HLA-E as an NK cell inhibitor [151]. This scaffold is compatible with allogeneic adoptive cell transfer, reduces the risk of GvHD and HvG rejection, and may also improve persistence and anti-tumor efficacy due to its low immunogenicity.

## 5. Production of CAR-T Cells

Although CAR-T cells have undergone multiple generations of development with the above-mentioned subtle modifications, the manufacturing process does not change much as outlined below (Figure 3). Firstly, monocytes are collected from the patient’s peripheral blood with leukapheresis. Secondly, specific T-cell subsets are isolated and achieved activation in a cell culture medium incorporating CD3/CD28 monoclonal antibody magnetic beads or artificial antigen-presenting cells. Thirdly, CAR expression on T cells is realized by lentivirus, gene editing, transposons, and other methods. Fourthly, T cells are expanded to the required level for the preparation of reinfusion. Although the manufacturing process of CAR-T cells is relatively mature and largely unchanged, there are still limitations such as time-consuming, expensive, and labor-intensive manufacturing. We describe the progress of CAR-T cells in terms of source, culture conditions, and manufacturing cycle, which is expected to greatly increase the popularity of CAR-T cells in clinical applications. We describe advances in the sources, culture conditions, and manufacturing cycles of CAR-T cells, which are expected to greatly improve the popularization of its clinical application.

Autologous CAR-T cells are produced using the patient’s own T cells. First, the patient’s white blood cells are collected through a process called leukapheresis. These collected cells are then sent to a specialized laboratory where the T cells are isolated and activated. The activated T cells are modified (lentivirus, retrovirus, RNAs, transposons) to express CARs that recognize and bind to the patient’s cancer cells. Finally, the CAR-T cells are expanded in culture to obtain a large enough population for infusion back into the patient. Allogeneic CAR-T cells are produced using T cells from a healthy donor and then these cells were collected similarly through leukapheresis and modified to express CARs. The following important step is to eliminate expression of αβ TCR on said T cells and CD52. These genetically modified T cells are then expanded in culture to produce multiple cells for infusion into a larger number of patients.

### 5.1. The Sources of CAR-T Cells

Currently, T cells used for CAR-T cell therapy are harvested predominately from the patient’s own T lymphocytes without the risk of an allogeneic reaction which is considered an advantage. However, patients heavily treated with a series of chemotherapies suffer a high tumor burden and T-cell deficiency, posting constriction on large-scale clinical applications and negatively impacting the quality of the CAR-T cells. In addition, such a patient-by-patient production platform requires costly and lengthy production translating into a delay in the availability of the CAR-T therapy, which also impedes the wide application for diverse tumor types. All these together contribute to the following attempts at allo-CAR-T cell therapy and universal CAR-T cells.

For CAR-T cells derived from suitable donors, preliminary results about donor-derived CAR19 T cells showed that regression of B-cell malignancies occurred in patients with malignancies refractory to standard donor lymphocyte infusion without GVHD in the absence of lymphocyte-depleting chemotherapy [33]. Moreover, research showed that allo-reactive T cells with the expression of CD28-costimulated CD19 CARs have gained better stimulation, resulting in the progressive loss of effector function and proliferative potential, thus significantly decreasing the occurrence of GVHD [143]. Nevertheless, there are still certain restrictions on donor-derived CAR-T cell treatment, including high costs, long manufacturing times, and constrained cell sources.

A promising development that may address most of these issues is to create a universal CAR-T (UCAR-T) cell therapy. It comprises allogeneic T cells from umbilical cord blood, peripheral blood, or regenerative stem cells. This approach frequently relies on gene editing to effectively delete the TCR gene or HLA class I locus of the allogeneic T cells, rendering the T cells insensitive to allogeneic antigens. A brand new immune-evasive universal CAR-T cells scaffold was recently described by Jo et al. using a combination of precise TALEN-mediated gene editing together with DNA matrices vectorized by recombinant adeno-associated virus 6 [151].

General-purpose universal CAR-T cells may overcome most of the limitations of autologous CAR-T cells, but the large-scale industrialization of CAR-T cells using allogeneic T lymphocytes as a source still awaits an attractive breakthrough. Currently, HSCs derived from cord blood or bone marrow are being tried to make CAR-HSCs, which in turn can be differentiated into effector cells, including CAR-T cells and CAR-NK cells. Boyd et al. developed a method to induce T cells from umbilical cord HSCs without co-culture stromal support lines [152], while Li et al. generated human allogeneic HSC-engineered iNKTs by combining HSCs genetic engineering and in vitro differentiation [153]. There are certain obstacles to the clinical translation of these methods, and the use of HSCs needs to face the challenge of the limited expansion capacity of primary cell banks while retaining their stemness. But its manufacturing feasibility, cancer treatment potential and high safety also provide a new possibility for anti-cancer immunotherapy.

By isolating and screening reprogrammed pluripotent stem cells (iPSC) in vitro, monoclonal clones with relatively high gene editing efficiency can be selected as a source of gene editing cell libraries. However, the differentiation systems for most iPSCs rely on the co-culture of mouse stroma-supported cells with low differentiation efficiency. Moreover, it has been shown that the molecular characteristics of iPSC-T cells are different from mature αβ T cells but resemble innate γδ T cells, thus affecting the immune function and antitumor activity. Recently, a breakthrough on this issue was reported showing that iPSCs-derived T cells could be produced through a combination of matrix-support cell-independent in vitro differentiation of iPSCs and EZH1 knockdown-mediated epigenetic reprogramming. This new in vitro differentiation system exhibited extremely high similarity to peripheral blood αβ T cells and demonstrated potent antitumor activity [154].

### 5.2. Culture Conditions for CAR-T Cells

Although evidence from clinical trials has demonstrated that CAR-T cell therapy can induce complete and long-lasting tumor regression in advanced cancer patients, response rates are still inadequate. There is now extensive consensus that objective responses are closely related to early T cell implantation levels, early amplification peaks, and the persistence of T cells, which are greatly influenced by the composition of the infused T cell product.

There are two subpopulations of memory T cells: peripheral tissue-homing effector memory T cells (T_EM_) and lymphatic tissue-homing central memory T cells (T_CM_). The “stem” of memory T cells, referred to as stem cell memory T cells (T_SCM_), has been previously discovered [155,156]. While naive cells, not T_CM_ cells, increased effector population mediating superior anti-tumor immunity upon adoptive transfer, some comparative studies showed tumor-reactive CD8+ T cell populations with T_CM_ phenotypic and functional attributes might be superior to T_EM_ regarding adoptive immunotherapies [157]. These T_SCM_s showed high proliferation and long-term survival, generating effector T cells with strong anti-tumor activity, which is a powerful weapon for adoptive cell transfer immunotherapy against cancer [155,158].

Requirements on cell culture conditions are vital in T_SCM_ conduction. Cieri et al. described the condition for the generation of the T_SCM_ subset as CD3/CD28 engagement and culture with IL-7 and IL-15 [159]. Overly intense co-stimulation, however, may induce differentiation of T_SCM_ to T_EM_. With such concern, Alvarez-Fernandez et al. conducted a research and found that, compared to a prolonged co-stimulation, a short anti-CD3/CD28 co-stimulation of naïve T cells, combined with IL-7 and IL-15, could greatly increase the frequencies of CD4+ and CD8+ T_SCM_ ex vivo [160]. Furthermore, the addition of IL-21 also enhanced the enrichment and amplification of CD4+ and CD8+ T_SCM_. Apart from cytokine induction, T_SCM_ cell generation can be achieved through activation of the Wnt-β-catenin signaling pathway by Gsk-3β inhibitors, where β-catenin deposited in the cytoplasm is transferred to the nucleus to bind to TCF-1/LEF1 transcription factors [161]. Further, compared with conventional CAR-T cells, both Notch-induced CAR-T_SCM_ cells and Foxm1-induced CAR-T_SCM_ cells both have stronger anti-tumor potential compared to conventional CAR-T cells [162].

Since CAR-T cells continuously differentiate with time, it is worthwhile to compare the early with late time points in the manufacturing process and draw the conclusion of the better ex vivo culture time. Studies showed that the number of adoptively transferred T_SCM_ in the product on Day 9 will be proportionally reduced compared to that in the Day 3 product [163]. In summary, after shortening the duration of T cell stimulation in vitro, T cells could enhance proliferation, secrete effector cytokines, and have better antitumor activity.

### 5.3. Manufacturing Cycle of CAR-T Cell

In general, a typical complete manufacturing cycle of CAR-T cells takes 2–4 weeks, with the common steps of T cell activation, viral transduction, and in vitro amplification. However, the activation and expansion of CAR-T cells can negatively affect therapeutic performance by contributing to the deterioration of differentiation and antitumor activity. Therefore, there is now a widely accepted consensus for shortening the preparation time and minimizing the cost to extend the accessibility of CAR-T therapy.

Ghassemi et al. reported a technique that could rapidly produce highly potent CAR-T cells within 24 h by adjusting the medium formulation and the surface area to volume ratio of the culture vessel [164]. The study exploited the unique ability of lentiviral vectors to enter and integrate into the genome of non-dividing cells, allowing direct lentiviral vector transduction of T cells without an activation step. In a mouse xenograft model of human leukemia, such rapidly generated non-activated CAR-T cells were demonstrated to display higher in vivo anti-leukemia activity than CAR-T cells generated following standard protocols, providing an efficient production of CAR-T cell products with durable engraftment capacity and efficacy. Meanwhile, researchers have developed an implant called MASTER (multifunctional alginate scaffold for T-cell engineering and release), a multifunctional alginate scaffold for T-cell modification and release [165]. After T cells were isolated from the body, inactivated T cells were mixed with a viral vector encapsulating the CAR gene and then poured onto the MASTER implant. The large pores and spongy nature of the MASTER material tightly bind the viral vector to the cells, which accelerated T-cell reprogramming. Moreover, accompanied by the infiltration of interleukin factors, all these together resulted in shorter differentiation and expansion time, facilitating better maintenance of the anti-cancer activity and duration of the cells. Based on this discovery, the team surgically implanted MASTER into a mouse lymphoma xenograft model, showing that CAR-T cells generated in vivo entered the circulation and controlled distal tumor growth, demonstrating greater persistence compared to conventional CAR-T.

In Jan 2022, Rurik et al. developed a novel therapeutic approach to generate transient antifibrotic CAR-T cells in vivo by delivering modified messenger RNA (mRNA) in T cell–targeted lipid nanoparticles (LNPs) [166]. Efficient delivery of modified mRNA encoding the CAR to T lymphocytes was observed, which produced transient, effective CAR-T cells in vivo. With this CAR-T design, the T cells in the human body are transformed into CAR-T cells without the need for in vitro production, thereby not only reducing the price of CAR-T cell therapy dramatically but also solving the problem of immune rejection.

## 6. Future Perspectives

### 6.1. Managing the Toxicity of CAR-T Cell Therapy

The toxicities associated with CAR-T cell therapy can be categorized into two main types: immune-related toxicities and target-related toxicities [1,2]. Immune-related toxicities primarily result from the activation of CAR-T cells, along with the subsequent release of excessive cytokines, including CRS and ICANS [167,168]. CRS is defined as a systemic disease induced by the overactivation of immune effector cells and massive amounts of various proinflammatory cytokines, including IL-1, IL-6, IFN-γ, and granulocyte-macrophage colony-stimulating factor (GM-CSF) [169,170]. Monocyte and macrophage lineages are also considered to be the key origin of inflammatory cytokines in relation to CRS [171]. Notably, the key cytokine for CRS is IL-6, which is the main cause of the immune reactions of CRS such as fever, chills, headache, and malaise. Likewise, it also gives rise to numerous potentially life-threatening symptoms such as vascular leakage syndrome and disseminated intravascular coagulation (DIC). Therefore, anti-inflammatory therapy, specially targeting IL-6, lays a solid foundation in the CRS management [172]. Tocilizumab, a humanized IgG1k anti-IL-6R antibody which inhibits the signal transduction pathways by binding to both soluble and membrane-bound IL-6R, has shown impressive efficacy in multiple clinical trials [173]. The therapeutic effects of ruxolitinib, an oral JAK1/2 inhibitor, have also been reported in steroid-refractory CRS [174]. ICANS stands for a complex condition involving the interaction between the infused CAR-T cells, the target tumor cells, and the central nervous system (CNS) [175]. The substantial release of inflammatory cytokines and chemokines during CAR-T cell therapy can penetrate the blood–brain barrier, inducing neuroinflammation and disturbing normal brain function [176]. Moreover, ICANS is characterized by endothelial dysfunction, blood–brain barrier compromise, and heightened permeability, leading to enhanced immune cell infiltration into the CNS and subsequent neurotoxic effects [177]. The severity of ICANS can range from mild symptoms like confusion and headaches to more severe manifestations such as seizures and cerebral edema. Prompt recognition and management are crucial to prevent long-term neurological damage [178]. Current strategies involve early intervention with supportive care, including corticosteroids to suppress inflammation, anti-seizure medications, and the close monitoring of neurological symptoms [179]. Research is ongoing to better understand the underlying mechanisms of ICANS and develop targeted therapies to mitigate its effects.

Target-related toxicity occurs because malignant cell antigens targeted by CAR-T are often also expressed at different levels in normal tissues, also known as on-target, off-tumor toxicity (OTOT) [180]. Therefore, the selection of target antigens for CAR-T cell therapy is critical, and the best antigen candidates should be expressed only on malignant cells to minimize the impact on normal tissues [181]. Current studies have achieved predictability of OTOT risk through preclinical models. Histological analysis and immunohistochemistry of mouse organs can verify extratumoral CAR-T cell infiltration, thus providing clear evidence for OTOT [182]. In addition, engineering strategies are being explored to improve the specificity of CAR-T cells and minimize off-target effects. This includes optimizing the design of CAR molecules, incorporating safety switches that can control CAR-T cell activity, and logic-gated CAR-T cells [183,184,185,186]. The close monitoring of patients receiving CAR-T cell therapy is essential to identify and manage any potential OTOT effects. The tyrosine kinase inhibitor dasatinib can achieve the reversible control of CAR-T cells. Although this inhibitor has not been tested clinically, it is expected to be an effective switch of CAR-T cell signaling [187,188].

### 6.2. Exploring beyond T Cells

CAR-T cells have shown remarkable success in cancer immunotherapy, but they are not the only type of immune cells being explored for this purpose. Other immune cells, such as CAR-NK cells, iNKT cells, and MAIT cells, are also being investigated for their potential in cancer therapy.

NK cells make up 5–15% of human peripheral blood leukocytes and are identified by the expression of CD56 without co-expression of CD3 and TCR [189]. CAR-NK cells, engineered from NK cells, have the potential to improve antitumor cytotoxicity by incorporating NK cell-specific signaling domains such as DAP-10, DAP-12, and 2B4 [190,191], while most CAR-NKs currently follow the structure of CARs in CAR-T therapies. CAR-NK cells have several advantages over CAR-T cell therapies, based on the unique characteristics of NK cells. These include a reduced risk for alloreactivity and GVHD even with allogeneic NK cells, which allows to generate CAR-NK cells from multiple sources such as NK92 cell lines, peripheral blood mononuclear cells (PBMCs), umbilical cord blood (UCB), and iPSCs, enabling large-scale production [192,193]. Additionally, NK cells produce a different spectrum of secreted cytokines from T cells, making NK cells producing IFN-γ and GM-CSF less likely to be associated with CRS and other severe neurotoxicity [34,194]. Furthermore, CAR-NK cells have the innate ability to exert cytotoxic activity not restricted to the CAR-dependent mechanism [195]. However, there are still challenges to overcome before the clinical application of CAR-NK cells. Major barriers for NK cell-based immunotherapy approaches include the lack of an efficient gene transfer method in primary NK cells, short-term persistence in the absence of relevant cytokines, and poor efficiency of homing to tumor sites [196,197,198]. Nonetheless, CAR-NK cells hold promise as a targeted cancer treatment option, and ongoing research may pave the way for more effective and efficient applications in the future.

Natural Killer T (NKT) cells are a unique non-MHC-restricted T lineage cell type. Among human NKT cells, the major subtype is invariant natural killer T (iNKT) cells, which can mature upon recognition of glycolipid antigens presented by CD1d molecules. iNKT cells possess potent anti-tumor activity through multiple mechanisms, including direct tumor lysis, modulation of cytokines in other immune effector cells, infiltration of tumors via various chemokines, and inhibition of immunosuppressive cells [199]. As such, iNKT cells are emerging as a promising alternative to CAR-based strategies. What sets iNKT cells apart is their TCR composition and antigen recognition pattern, which lack any potential for inducing GvHD, and may even help prevent it [200]. This makes them an attractive option for universal CAR clinical studies. However, several obstacles must be overcome before iNKT cell-based therapies become widely accessible. The low frequency of NKT cells in human peripheral blood mononuclear cells, along with the difficulty in expanding and culturing them in vitro at clinical scale, necessitates the development of new methods to meet the required numbers [201]. Additionally, CAR-iNKT cells exhibit short-term persistence, and repeated doses or additional cytokine administration may be required to achieve sustained therapeutic effects. Therefore, optimizing CAR strategies is essential to expand their use and improve long-term efficacy.

Mucosa-associated invariant T cells (MAIT) are unconventional T cells that recognize microbial-derived riboflavin derivatives presented by the conserved MR1 molecule and are endowed with powerful effector functions [202]. Because they are not selected by the classical MHC/peptide complex and express a semi-invariant T-cell receptor, MAIT cells do not mediate allogeneic responses, prompting their use as a novel source of general-purpose effector cells for allogeneic CAR-T cell therapy without the need to inactivate endogenous TCRs. In addition, MAIT cells recognize microbial peptides presented by the highly conserved MR1 and thus connecting the innate and acquired immune systems, thereby mediating an enhanced immune response. Given the representative MAIT cell ligand 5-OP-RU, MAIT cells can be efficiently activated and show enhanced antitumor capacity [203]. However, the isolation, expansion and transfer of MAIT cells back into patients using current methods remains a challenge [204]. Due to the limited number of MAIT cells in individuals, the number of CAR-MAIT obtained after expansion will also be limited compared to the in vitro expansion of all T cells. The PD-1 upregulation and depletion phenotype of MAIT cells has been observed in chronic viral hepatitis, suggesting that a combination with anti-PD1 may overcome this barrier [205].

### 6.3. Challenges in Solid Tumors and Other Diseases

With the rapid progress of our knowledge about diseases and their underlying molecular mechanisms, many molecules have been identified as therapeutic targets, making immune cell therapy an attractive and promising therapeutic option for more diseases other than hematological tumors. Translation of the currently achieved success in CAR-T cell therapy for hematological malignancies to treat solid tumors remains challenging. Currently, major obstacles for CAR-T cell treatment to cure solid tumors lie mainly in choosing the appropriate targets and overcoming the tumor microenvironment. Up to now, available target antigens for solid tumors mainly comprise HER2, EGFR, mesothelin, NY-ESO-1, and PSMA [53,68,206,207,208]. Among them, NY-ESO-1 is expressed only in testicular or ovarian tissues but re-expressed in many cancers, which makes it an ideal target antigen with strong specificity and weak side effects for potential CAR-T-cell therapy. In addition, a study reported that 12 children with relapsed/refractory neuroblastoma showed positive clinical anti-tumor activity after treatment with GD2-targeted CAR-T cells. The treatment was well tolerated without reported off-target toxicity [209].

A further strategy for enhancing the specificity of CAR-T treatments is to engineer receptors to improve their ability to recognize tumor cells. By designing a two-step positive feedback circuit, researchers addressed the carnage of CAR-T cell therapy on normal cells with decreased tumor antigens expression that enabled killer T cells to differentiate targets based on S-type antigen density thresholds [110]. In HER2-expressing cancer cells and mouse tumor models, the technique can precisely and efficiently kill cancer cells with elevated HER2 expression but spare normal cells with reduced HER2 expression. Additionally, implementing “off switches” can help CAR-T cells improve specificity and reduce toxic effects when treating solid tumors. For example, the synNotch system can not only accurately aggregate to glioblastoma but also detect and kill other cancer cells in the surrounding area, providing new research ideas for treating a variety of cancers that have long been untreatable with traditional CAR-T [210].

Targets that can be utilized in CAR-T cell treatment are emerging constantly with research going on. CAR-T cell therapy has attracted growing scientific and clinical interest due to its significant and sometimes revolutionary success in oncology treatment. In terms of autoimmune disease, standard methods include targeting antibody-producing cells, replacing scFv with recombinant autoantigens, and introducing CARs into Treg cells. A case report of a treatment-resistant SLE patient who was successfully treated with CAR-T therapy published last year has directed a new direction of treatment strategies for patients with autoimmune disorders [211]. As for infectious diseases, researchers have designed a CAR-T cell capable of recognizing hepatitis B virus surface antigen (HbsAg) [212]. The findings illustrated that HBsAg-redirected T cells exhibited antiviral activity in HBV-infected human liver chimeric mice. In addition, for allergic diseases with IgE-mediated type IV hypersensitivity reactions, clearance of IgE and IgE-secreting B cells and plasma cells can be achieved with anti-IgE CAR-T cells [213]. Furthermore, by constructing anti-fibroblast activation protein (FAP) CAR-T cells, the activity of fibroblasts can be inhibited, thus reducing or even reversing cardiac fibrosis in mouse models [214]. All these above-mentioned findings would help us enter a new era, in which CAR-T therapies are not only used for malignancies but also helpful for other disorders. 

## 7. Conclusions

This article provides an insightful overview of the recent advancements in CAR-T cell structure, production, and the strategies employed to mitigate toxic side effects. The subtle modifications to the CAR structures from generation to generation, as well as progress in the production of CAR-T cells would pave the way towards the ultimate goal of cure for various diseases.

## Figures and Tables

**Figure 1 cancers-15-03476-f001:**
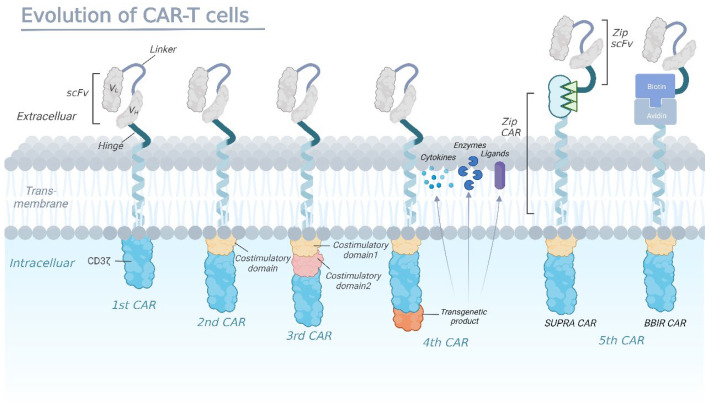
The evolution of chimeric antigen receptors (CAR)-T cells.

**Figure 2 cancers-15-03476-f002:**
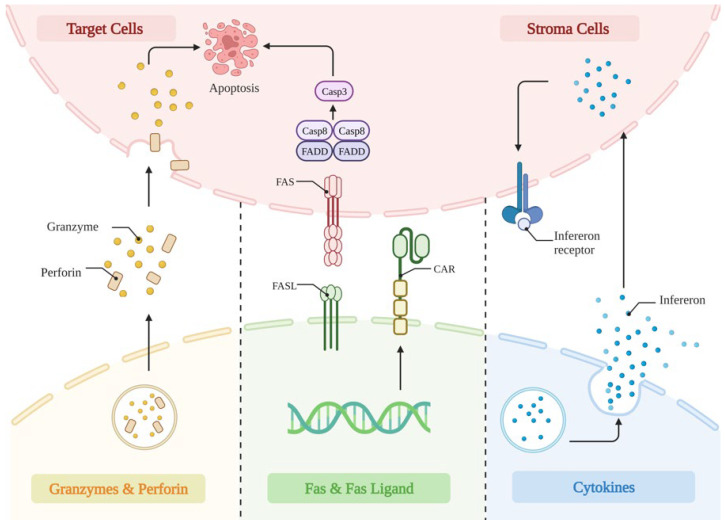
The anti-tumor mechanism of CAR-T cells. (1) Perforin functions by drilling holes in the membranes of tumor cells, through which granzymes are delivered into the tumor cells, resulting in apoptosis. (2) CAR-T cells highly express tumor FAS ligands on their surface, further inducing apoptosis in tumor cells. (3) The released cytokines by CAR-T cells modulate tumor lysis through upregulating interferon on stroma cells, alter the tumor microenvironment, and further enhance its anti-tumor activity.

**Figure 3 cancers-15-03476-f003:**
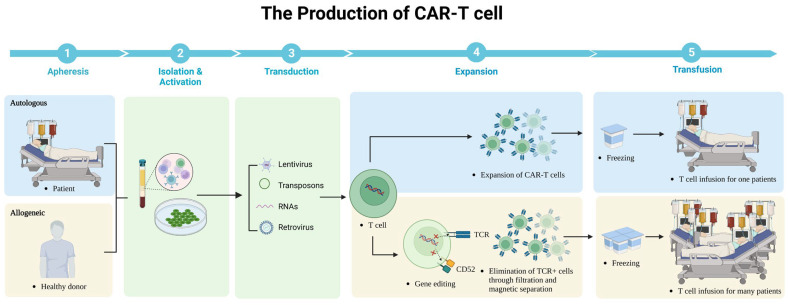
The production of CAR-T cells.

**Table 1 cancers-15-03476-t001:** FDA-approved CAR-T cell therapies.

Production (Target)	Approval Date	Indication	Pivotal Study
Kymriah^®^ (CD19)	August 2017	Patients up to 25 years of age with B-cell ALL that is refractory or in second or later relapse	ELIANA trial [42]
	May 2018	Adult patients with r/r LBCL after two or more lines of systemic therapy, including DLBCL-NOS, HGBCL and DLBCL arising from FL.	JULIET trial [43]
	May 2022	Adult patients with r/r FL after two or more lines of systemic therapy	ELARA trial [44]
Yescarta^®^ (CD19)	October 2017	Adult patients with r/r LBCL (including PMBCL, HGBCL and DLBCL arising from FL) after two or more lines of systemic therapy	ZUMA-1 trial [1]
	March 2021	Adult patients with r/r FL after two or more lines of systemic therapy	ZUMA-5 trial [45]
	April 2022	Patients with r/r LBCL and no more than 12 months after first-line chemoimmunotherapy	ZUMA-7 trial [46]
Tecartus^®^ (CD19)	July 2020	Adult patients with r/r mantle cell lymphoma	ZUMA-2 trial [47]
	October 2021	Adult patients with r/r B-cell precursor ALL	ZUMA-3 trial [48]
Breyanzi^®^ (CD19)	February 2021	Adult patients with r/r LBCL who have previously received 2 or more systemic therapies	TRANSCEND NHL 001 trial [49]
	June 2022	Adult patients with r/r LBCL: - Refractory to first-line chemoimmunotherapy; - Relapse within 12 months after first-line chemoimmunotherapy; - Relapse after first-line chemoimmunotherapy and unsuitable for HSCT due to comorbidities or age.	TRANSFORM trial [50]
ABECMA^®^ (BCMA)	March 2021	Patients with r/r multiple myeloma after at least three previous regimens including a proteasome inhibitor, an immunomodulatory agent, and an anti-CD38 antibody.	KarMMa trial [51]
CARVYKTI^®^ (BCMA)	February 2022	Patients with multiple myeloma: - Received three or more previous lines of therapy; - Double-refractory to a proteasome inhibitor and an immunomodulatory drug, and had received a proteasome inhibitor, immunomodulatory drug, and anti-CD38 antibody.	CARTITUDE-1 trial [52]

ALL, acute lymphoblastic leukemia; r/r, relapse/refractory; LBCL, large B-cell lymphoma; DLBCL-NOS, diffuse large B-cell lymphoma-not otherwise specified; HGBCL, high-grade B-cell lymphoma; FL, follicular lymphoma; PMBCL, primary mediastinal B-cell lymphoma; HSCT, hematopoietic stem cell transplantation.

**Table 2 cancers-15-03476-t002:** CAR-T cell products in preclinical and early-phrase clinical studies.

CAR-T Cell Product	Targeted Diseases
Mesothelin-CAR-T cell	Mesothelioma [53], ovarian cancer [54], pancreatic cancer [55], lung cancer [56]
GD2-CAR-T cell	Neuroblastoma [57], melanoma [58], osteosarcoma [59], retinoblastoma [60], GD2-positive breast cancer [61], lung cancer [62]
ROR1-CAR-T cell	Lung cancer [63], triple-negative breast cancer [63,64]
HER2-CAR-T cell	Progressive HER2-positive glioblastoma [65] uveal and cutaneous melanoma [66], metastatic colorectal cancer [67]
NY-ESO-1-CAR-T cell	Triple-negative breast cancer model and primary melanoma tumor [68]
A. fumigatus-CAR-T cell	Antifungal reactivity in preclinical models in vitro and in vivo [69]
CD70-CAR-T cell	Renal cell carcinoma [70], CD70-positive AML (without HSC toxicity) [71], gliomas [72], CD19-negative B-cell Lymphoma [73]
AXL-specific CAR-T cell	Non-small cell cancer [74], triple negative breast cancer [75]
FRα-CAR-T cell	Breast cancer [76], ovarian cancer [77]
MUC1-CAR-T cell	Esophageal cancer [78], a head and neck squamous cell carcinoma [79], cholangiocarcinoma [80]
PSMA-CAR-T cell	Prostate cancer [81,82]
CD33-CAR-T cell	Acute myeloid leukemia [83]
CD44v6-CAR-T cell	Lung and ovary adenocarcinoma [84], AML [85]

Chimeric antigen receptor, CAR; Receptor tyrosine kinase-like orphan receptor 1, ROR1; prostate specific membrane antigen, PSMA; Folate receptor alpha, FRα.

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
