# Peer review of "Fine-Tuning through Generations: Advances in Structure and Production of CAR-T Therapy"

_cancers, 2023, doi:10.3390/cancers15133476_

Round 1
Reviewer 1 Report
In this Review, Zheng et al. summarized the history and current progress of CAR-T cell therapy. Lots of papers have reviewed the similar topic and I would suggest that authors highlight the genetic and manufacturing modifications which enhance the efficacy, safety and persistence of CAR-T therapy. I have some suggestions here:
1. In the title, it should be “make” instead of “makes”
2. Authors should add another table to present other CAR-T cell products in preclinical studies, such as mesothelin-CAR, GD2-CAR and others.
3. More genetic modifications should be mentioned, such as knock-out of B2M and CIITA for reduced immunogenicity, overexpression of HLA-E for reduced allo-NK rejection, and others.
4. Besides CAR-T cells, authors should briefly talk about CAR-NK, iNKT, MAIT and other immune cells for cancer therapy.
5. Besides iPSCs, HSCs are another important source for the generation of CAR-T, CAR-iNKT, and CAR-MAIT cells. Please discuss the importance of HSCs.
6. Figure 2 only shows the production of autologous CAR-T cells, allogeneic off-the-shelf CAR-T cell therapy should be presented as well.
7. CAR-T cell-induced CRS and other side effects cause serious outcomes for cancer patients. Are there efficient strategies and modifications to reduce the side effects?
8. Authors could briefly introduce the strategies to overcome antigen escape of CAR-T cells.
Reviewer 2 Report
Comments to authors
The overall study is interesting and suitable for publication after considering minor revisions.
The abstract section looks short. The authors should add more background of the study.
Please rephrase the aims and objectives of the study and rewrite them with more details.
Authors should address the gaps present in this field with reference to previous studies in the introduction section.
Figure 1 is confusing and doesn't explain the evolution. The authors should add some more information/figures to explain the evolution of CAT-T cells.
In table 1 there is no reference citation and the header is too short to explain the major objectives of the data in the table.
Support your opinion with more recent references.
Be consistent with your style while writing the units and symbols.
Add more details to the figure legends. They must be self-explanatory.
Replace the old references with the studies of the last 10 years. You can add the citations of the older studies only in case of special circumstances.
Round 2
Reviewer 1 Report
No further comment
Author Response
Thank you very much for your precious comments.